# Identifying Partial Mouse Brain Microscopy Images from the Allen Reference Atlas using a Contrastively Learned Semantic Space

Justinas Antanavicius[1], Roberto Leiras[2], Raghavendra Selvan[1,2]

[1] Department of Computer Science, University of Copenhagen, Denmark
[2] Department of Neuroscience, University of Copenhagen, Denmark
`raghav@di.ku.dk`

**Abstract.** Registering mouse brain microscopy images to a reference atlas is crucial to determine the locations of anatomical structures in the brain, which is an essential step for understanding the function of brain circuits. Most existing registration pipelines assume the identity of the reference plate – to which the image slice is to be registered – is known beforehand. This might not always be the case due to three main challenges in microscopy image data: missing image regions (partial data), different cutting angles compared to the atlas plates and a large number of high-resolution images to be identified. Manual identification of reference plates as an initial step requires highly experienced personnel and can be biased, tedious and resource intensive. On the other hand, registering images to all atlas plates can be slow, limiting the application of automated registration methods when dealing with high-resolution image data. This work proposes to perform the image identification by learning a *low-dimensional* space that captures the similarity between microscopy images and the reference atlas plates. We employ Convolutional Neural Networks (CNNs), in the *Siamese* network configuration, to first obtain low-dimensional embeddings of microscopy image data and atlas plates. These embeddings are contrasted with positive and negative examples in order to learn a semantically meaningful space that can be used for identifying corresponding 2D atlas plates. At inference, atlas plates that are closest to the microscopy image data in the learned embedding space are presented as candidates for registration. Our method achieved TOP-3 and TOP-5 accuracy of 83.3% and 100%, respectively, compared to the SimpleElastix-based baseline which obtained 25% in both the Top-3 and Top-5 accuracy.[3]

**Keywords:** Image registration · Mouse brain · Partial data · Deep Learning.

## 1 Introduction

Determining the location of anatomical structures in a mouse brain is an essential step for analyzing and understanding the architecture and function of brain circuits, and of the overall whole-brain activity [3]. Structures of interest can be located using standardized anatomical reference atlases, usually taking a two-step approach:
**1. Identification:** The input brain slice has to be identified, i.e., the corresponding 2D atlas plate has to be found.

---

[3] Source code is available at https://anonymous.4open.science/r/8A32/README.md

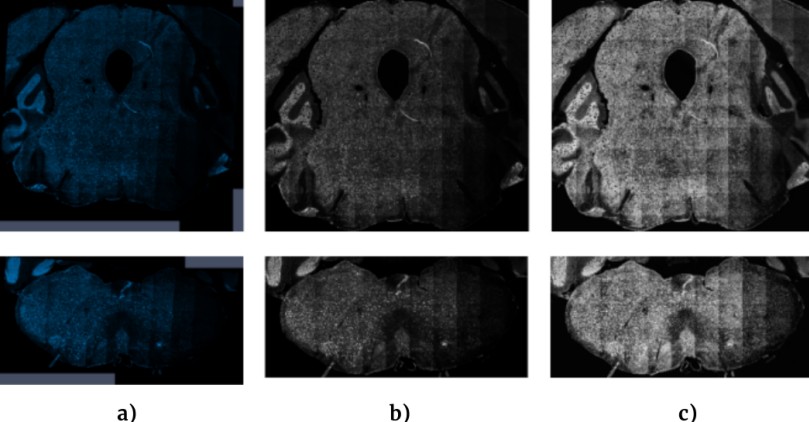

**Fig. 1.** a) Two typical high-resolution microscopy images showing the cross-sectional view of a mouse spinal cord in pseudo-color. The size of the input images in this work varied between 17408×10240 and 25600×20480 pixels. Notice the artefacts due to low contrast, tiling and missing regions, which make them challenging to process. b) Input images after gray scale conversion c) Pre-processed images with histogram equalization

**2. Registration:** The identified slice is registered to the corresponding atlas plate. Anatomical structures are determined based on the registered annotated plate.

In most cases, the acquired microscopy images of brain slices often suffer from artefacts due to missing tissue regions, irregular staining, titling errors, air bubbles and tissue wrinkles [12], as shown in Figure 1. This is further aggravated due to additional variations in the images depending on the experimental procedures, instrumentation noise, etc. This makes it difficult to identify and register mouse brain images. For these reasons, practitioners usually resort to manually comparing image slices to 2D atlas plates which can be very time-consuming.

Compared to the registration of mouse brain images, the first part of identification has received far less attention from the research community. At the outset, wrong identification of brain slices could lead to incorrect determination of anatomical structures regardless of how well the image registration itself is performed. Therefore, precise determination of anatomical structures requires accurate identification of brain slices as a precursor.

The correspondence between brain slices and atlas plates could be found by reconstructing a 3D volume from the brain slices and then registering them to the 3D reference atlas [10]. However, it is not always possible to construct an accurate brain volume, e.g. when brain slices are cut at different angles or when only few brain slices are available or partial brain images are used. The difference in slice cutting angles between the atlas plates and the acquired images is a common challenge affecting the usefulness of atlas-based registration. In Figure 2 we illustrate an instance where different regions of the same image could correspond to different atlas plates due to a mismatch between the cutting angle of the acquired image from a brain slice and the atlas plates. This way, the central region of an image corresponds to an atlas plate

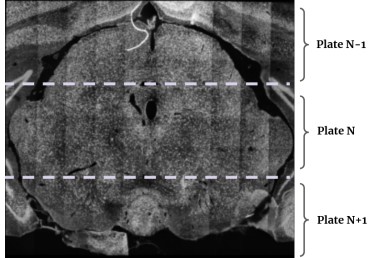

**Fig. 2.** Due to the difference in cutting angles compared to the atlas plates, no single ground truth plate can be registered to the input images. In this illustration, we point this out where the expert user usually would register different, usually consecutive, plates to different regions of the image.

(Plate N) while the upper part of the image belongs to the previous plate and the bottom part to the next atlas plate.

In this study, we investigate the problem of identifying the atlas plates corresponding to mouse brain slices, when the image data are partial and/or acquired at different cutting angles. The brain slices in this work are identified by finding the corresponding 2D coronal plates in the Allen Mouse Brain Atlas [8]. This is achieved by using convolutional neural networks (CNNs), used in the Siamese Network configuration [1, 7], to obtain low-dimensional representations of the image data. These low-dimensional embeddings are contrasted with positive and negative pairs to learn a semantically meaningful space where the correspondence between brain slices and atlas plates can be determined. The image identification method is compared to SimpleElastix, which is based on the widely used tool Elastix [9], in terms of accuracy and speed.

## 2   Methods

**Siamese Networks:** In this work, CNNs are used to identify brain slices by matching them to their corresponding atlas plates. The network architecture is comprised of identical CNNs in the Siamese Network configuration [7], as shown in Figure 3. The CNN, $S_\theta(\cdot)$, takes an image $I$ (of height H and width W) as input and outputs a low-dimensional feature vector (embedding), $h$, i.e., $S_\theta(\cdot) : I \in \mathbb{R}^{H \times W} \mapsto h \in \mathbb{R}^L$, where $L$ is the size of the embedding space and $\theta$ are the learnable network parameters. In the pairwise setting, two *sister* neural networks with shared parameters are used (Figure 3-b).

The embeddings for brain slices, treated as the fixed image, are obtained as $h_F = S_\theta(I_F) \in \mathbb{R}^L$. The embeddings for the atlas plates, treated as the moving image, are obtained in a similar manner, $h_M = S_\theta(I_M) \in \mathbb{R}^L$. After obtaining the embeddings of the fixed and moving images, their similarity is determined based on the Euclidean distance between these embeddings, $d(h_M, h_F)$. The reference atlas plate with the lowest distance is then predicted to be the corresponding atlas plate for a given brain slice.

**Metric learning:** The distance between the embeddings of more similar images should be smaller than that between dissimilar images for the low-dimensional embedding

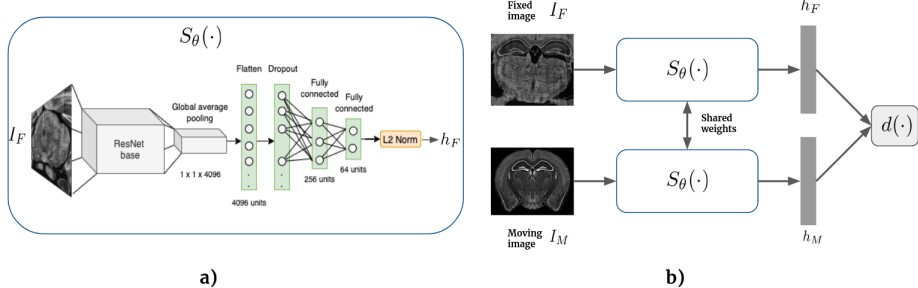

**Fig. 3.** a) Network architecture of the model used comprising of a ResNet-backbone and a multi-layered perceptron, $S_\theta(\cdot)$, used to obtain low-dimensional embeddings of the brain slices and atlas plates. b) Computing the similarity between brain slices and atlas plates with CNNs based on the low-dimensional representations corresponding to the moving and fixed images obtained from the identical CNNs, in a Siamese network layout, which are further used to compute their pairwise similarity, $d(\cdot)$.

space to be meaningful. This is achieved in this work using weakly supervised metric learning [2]. The Siamese networks for brain slice identification are trained to learn the representation of images such that corresponding brain slices and atlas plates would be closer to each other in the embedding space. We compare the embedding space learned based on training the networks with two different loss functions:
1. Contrastive loss [4], given as:

$$L = \begin{cases} \dfrac{1}{2}d(h_F,h_M)^2, & \text{if positive pair} \\ \dfrac{1}{2}\max(0,m-d(h_F,h_M))^2, & \text{if negative pair} \end{cases} \tag{1}$$

where the positive pair is comprised of the microscopy image, $I_F$, and the corresponding ground truth atlas plate, $I_M$, and the negative pair can consist of any non-ground truth atlas plate. The parameter $m \in \mathbb{R}_+$ is the margin used to control the contribution from negative pairs.
2. Triplet loss [11], given as:

$$L = \max(d(h_A,h_P) - d(h_A,h_N) + m, 0) \tag{2}$$

where $h_A$, $h_P$, $h_N$ are the embeddings of anchor- ($I_A$), positive- ($I_P$) and negative- ($I_N$) images, respectively. Note that in case of triplet loss, a third sister network with shared weights is included to obtain feature embeddings.

Two different types of triplets ($I_A$, $I_P$, $I_N$) are sampled to calculate the triplet loss. These triplets are defined based on the distance between the embeddings $h_A$, $h_P$, $h_N$ of anchor $I_A$, positive $I_P$ and negative $I_N$ images:

i) Semi-hard triplets: the distance between $h_A$ and $h_P$ is smaller than the distance between $h_A$ and $h_N$, however, the loss is still positive.
ii) Hard triplets: the distance between $h_A$ and $h_N$ is smaller than the distance between $h_A$ and $h_P$.

When the models are either trained with contrastive- or triplet- losses, the training process enforces structure to the embedding space so that the embeddings of similar images are pulled closer, whereas embeddings of dissimilar images are pushed away from each other. At inference, new microscopy images are ideally closer to their corresponding atlas plates in the embedding space. An overview of CNNs in Siamese network configuration for atlas plate prediction with moving and fixed images is shown in Fig. 3.

## 3   Data & Experiments

### 3.1   Data

**Microscopy data:** Eighty-four high-resolution microscopy images of mouse brain slices were acquired using a 10x objective in a Zeiss LSM 900 confocal microscope from four animals. The size of the images varied between $17408 \times 10240$px and $25600 \times 20480$ px. Most of the images were partial as they were not capturing the entire brain slice. For instance, the cortex or the cerebellar cortex were captured partially or, in some images, were not captured at all as seen in first column of Fig. 4. The images of brain slices were preprocessed, cropped and equalized using Contrast Limited Adaptive Histogram Equalization (CLAHE) to reduce some artefacts, as shown in Figure 1. The dataset was split into four sets: training (50 images), validation-1 (12 images), validation-2 (10 images) and test (12 images).

**Ground truth:** The Allen Mouse Brain Atlas [8] was used as the reference atlas. It consisted of 132 Nissl-stained coronal plates spaced at 100 $\mu$m, seen in the second column of Fig. 4. The ground truth in these experiments were the atlas plate numbers which were provided by a neuroscientist with expertise in manual registration of these images. For a given brain slice, there could be several matching plates due to the difference in cutting angles, as shown in Figure 2. However, the domain expert marked a single plate to be the ground truth depending on whichever plate best described specific regions of interest. This is to say, in most applications involving these data there are no hard ground truths as each slice could correspond to several consecutive atlas plates due to the difference in cutting angles.

**Data augmentation:** To capture variations in the microscopy data beyond the limited training set extensive data augmentation (affine transformation, cropping and padding, pepper noise) was applied to the training dataset. Data augmentation was performed on all the 50 training set brain slices and also the 132 atlas plates. In order to reduce computations, the high resolution images were resized to square inputs of size $1024^2$, $512^2$ or $224^2$ depending on the experiment.

### 3.2   Experiments

**Experiments**: The performance of our CNN-based slice identification method was compared with a baseline SimpleElastix-based algorithm that identifies brain slices based on mutual information (MI). The baseline method affinely registers each brain slice with every atlas plate and picks the atlas plate with the highest MI. In total, 100 random hyperparameters from the SimpleElastix affine parameter map were tested. The results of the best performing baseline model (with 7 resolutions using recursive image pyramid and random sample region, 2800 iterations in each resolution level

**Table 1.** Mean Absolute Error (MAE) on the *validation-2* dataset for identifying brain slices with our method. The lowest MAE is achieved by the network with ResNet50v2 base, trained with semi-hard triplet loss and using $1024^2$ images. B is the training batch size.

| Loss | B | ResNet50v2 $224^2$ | $448^2$ | $1024^2$ | ResNet101v2 $224^2$ | $448^2$ | $1024^2$ |
|---|---|---|---|---|---|---|---|
| **Triplet (semi-hard)** | 32 | 2.5 | 2.2 | 2.8 | 1.9 | 3.1 | 3.1 |
| | 16 | 2.0 | 3.7 | **1.8** | 2.6 | 2.1 | 2.7 |
| **Triplet (hard)** | 32 | 2.4 | 3.0 | 3.0 | 2.8 | 3.7 | 2.7 |
| | 16 | 3.1 | 2.8 | 2.6 | 2.0 | 2.7 | 2.8 |
| **Contrastive** | 32 | 3.6 | 2.1 | 3.4 | 4.2 | 2.5 | 5.6 |

and disabled automatic parameter estimation) are used for comparison.

**Metrics:** The methods were evaluated based on three metrics: Mean Absolute Error (MAE), TOP-N accuracy and inference time. MAE measured the accuracy of predictions. For each brain slice all 132 atlas plates were ranked (starting from zero) based on the similarity score (the Euclidean distance or MI, depending on the method). Then MAE was computed as $MAE = (\sum_{i=0}^{N} y_i)/N$, where N is the number of brain slices, $y_i$ is the position of ranked ground truth atlas plate for a given brain slice $i$. With 132 atlas plates used, MAE can have values in the range [0,131]. If all brain slices are identified correctly, MAE is equal to 0. To account for the inherent ambiguity in ground truth we report Top-3, Top-5 and Top-10 accuracy.

**Hyperparameters:** Fig. 3-a) shows the architecture of the Siamese Networks with the embedding space feature dimension $L = 64$. The base of network consists of a CNN-backbone implemented as ResNet network [5] pre-trained on the ImageNet dataset. The CNN backbone is followed by a multi-layered perceptron that outputs the embedding. While training the networks, all layers of the ResNets were *frozen* except the last ones starting with the prefix *conv5*. The networks were trained on the *training* dataset for a maximum of 10k iterations using the Adam optimizer [6] with an initial learning rate of $10^{-4}$. The experiments were performed on Nvidia GeForce RTX 3090 GPU, i7-10700F CPU and 32 GB memory. The training was stopped if MAE on the *validation-1* dataset was not decreasing for more than 2k iterations.

**Results:** The converged models based on *validation-1* set were evaluated on the *validation-2* dataset, and the MAE performance for two ResNet backbones (ResNet50, ResNet101), the various loss functions, input- and batch- sizes are reported in Table 1. The best performing configuration is the ResNet50 backbone network trained with batch size (B) of 16 using input size $1024^2$ with the semi-hard triplet loss with MAE=1.8. This best performing model was further evaluated on the *test* dataset and compared with the SimpleElastix-based approach, reported in Table 2. We notice that the MAE on test set for our method is 1.42 compared to 60.4 for the baseline. Our method obtained Top-3 accuracy of obtaining 83.3% compared to 25% for the baseline. A similar trend is observed for Top-5 and Top-10 accuracy, where our method achieves 100% accuracy. The total inference time on the *test* set for the two methods are also reported in Table 2 where we observe that the baseline method takes orders of magnitude more time than the trained CNN model.

Finally, the Top-5 predicted atlas plates on a subset of the *test* dataset are reported in Table 3. In all the cases, the ground truth plate is within the Top-5 predictions

**Table 2.** Performance of our CNN-based method compared to the SimpleElastix-based approach on the *test* dataset for identifying brain slices reported as Top-N accuracy. Our method trained with semi-hard triplet loss outperforms SimpleElastix-based approach by a large margin in all the evaluated metrics. Inference time measures the time taken to identify all 12 brain slices from the *test* dataset.

|  | MAE | TOP-1 | TOP-3 | TOP-5 | TOP-10 | Infer. time |
|---|---|---|---|---|---|---|
| SimpleElastix | 60.4 | 16.7% | 25% | 25% | 25% | 12h 25 m |
| Siamese Networks | **1.42** | **25%** | **83.3%** | **100%** | **100%** | **7.2 sec** |

**Table 3.** Identifying brain slices from the subset of the *test* dataset: the labels of ground truth and Top-5 predicted atlas plates by our CNN-based method. Even though some predictions are incorrect, all of them are close to the ground truth labels. Labels define the position of atlas plates in the reference atlas.

| Ground Truth | Top-5 predictions |
|---|---|
| 91 | 92, **91**, 93, 90, 94 |
| 130 | 129, 128, **130**, 131, 127 |
| 86 | 87, 88, **86**, 85, 89 |
| 63 | 62, 61, 60, **63**, 59 |
| 108 | 109, 110, 111, 112, **108** |

highlighted in bold. Examples of the predicted atlas plates by our method that have the highest similarity are visualized in Fig. 4.

## 4   Discussions & Conclusions

Our CNN-based method in the Siamese network configuration used to identify brain slices have shown impressive results, i.e. in finding corresponding coronal 2D atlas plates. Our method performed well even when most images were missing image regions, and some images belonging to different classes (plate numbers) looked very similar to each other, thus making the identification task even more complex. Training with contrastive- and triplet- losses solve this issue by using margin, i.e., dissimilar images are not pushed away if the distance between them is larger than the margin.

The identification accuracy (MAE) had no clear correlation with the batch size (16 and 32), the image resolution ($224 \times 224$, $448 \times 448$, $1024 \times 1024$) and the type of the base for the Siamese network (ResNet50v2 and ResNet101v2), as seen in Table 1. However, using images with lower resolution and networks with fewer parameters could further improve the inference time. We did not observe the performance of our method to be highly influenced by the choice of loss functions. The models trained with triplet loss rather than contrastive loss, on average, achieved higher accuracy, however, the difference is not significant.

Evaluating the performance of the method using ambiguous ground truth data due to variations in cutting angle was another challenge. This was overcome by evaluating the methods using Top-N accuracy instead of only predicting the most similar atlas plate. We observe that our method achieved TOP-5 accuracy of 100% meaning that the actual corresponding atlas plate always falls in the top 5 predicted atlas plates, as seen in Table 2. Further, the variations within the Top-5 predictions for all five cases reported in Table 3 could be plausible, as most of the predictions are neighbouring atlas plates of the ground truth. We also report the Top-1 accuracy and

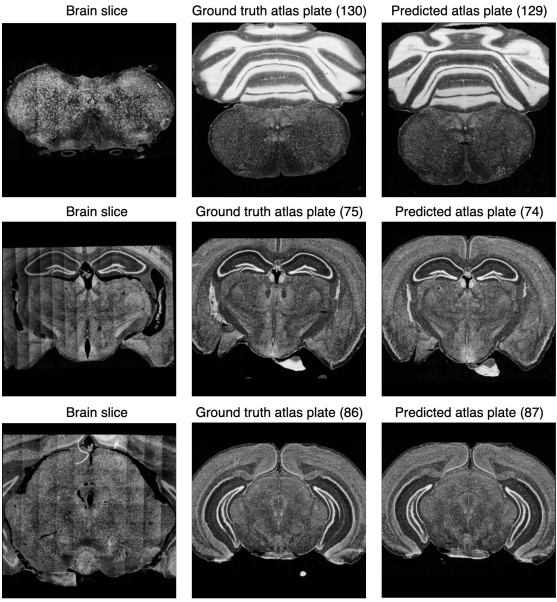

**Fig. 4.** Examples of the predicted (most similar) atlas plates by our method. Note that in all cases the ground truth plates are predicted within the Top-5 candidates in Table 3. Columns: **(1)** brain slices from the test dataset; **(2)** ground truth atlas plates; **(3)** predicted atlas plates. The number in parentheses shows the label of the atlas plate, i.e. the position of the atlas plate in the reference atlas.

notice a drop in performance for both methods due to the inherent ambiguity in the ground truth. The inherent ambiguity of the ground truth makes our method more useful as practitioners can explore several likely candidate atlas plates to register to.

In conclusion, we proposed to use CNNs in Siamese Network configuration trained with contrastive- and triplet- losses as a method for identifying correspondence between complete and partial mice brain slices. Challenges such as partial/missing data and variations in cutting angles were overcome by learning a semantically meaningful embedding space. Our method has shown large performance improvements in both accuracy and inference times compared to the SimpleElastix-based baseline. With this work, we have we demonstrated the usefulness of this approach with a 2D reference atlas. We hypothesize that the same method can also be applied to a 3D reference atlas for further improved precision in the slice identification task.

**Acknowledgments** The authors thank Kiehn Lab (University of Copenhagen, Denmark) for providing access to the microscopy images and the hardware used to train the models. They also acknowledge the Core Facility for Integrated Microscopy (CFIM) at the Faculty of Health and Medical Sciences for support with image acquisition.

**Compliance with Ethical Standards**: All animal experiments and procedures were carried according to the EU Directive 2010/63/EU and approved by the Danish

Animal Experiments Inspectorate (Dyreforsøgstilsynet) and the Local Ethics Committee at the University of Copenhagen.

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
