# OpenReview forum: "Identifying Partial Mouse Brain Microscopy Images from the Allen Reference Atlas using a Contrastively Learned Semantic Space"
_WBIR.info/2022/Workshop/Biomedical_Imaging_Registration — WBIR 2022_

### Official Review · Reviewer_FWz9 · 2022-02-14

**Rating:** 4
**Confidence:** 4

**Deanonymize Review:**

no

**Detailed Comments:**

1. How much do the results depend on using data augmentation?
2. What would a baseline performance for an embedding obtained from a pre-trained network be? I.e., does the metric learning step improve results significantly or would embeddings from a pre-trained network already be pretty good for ranking? This would be of particular interest as the conclusion is that neither batch size, nor loss function, nor image resolution, nor type of Siamese network base architecture had large effects on performance.
3. How resilient to artifacts is the proposed approach expected to be? And how difficult would it be to train a system that could also handle tilting artifacts? Would this be possible with a 3D atlas?
4. As a naive baseline approach one could simply treat this problem as a classification problem (with plate number as class). Would such an approach be obviously inferior to the embedding strategy?


**Paper Type:**

validation / application paper

**Strengths Weaknesses:**

This work makes use of metric learning via a Siamese network to learn embeddings for 2D microscopy images so that they can be matched with the plates of an atlas. The Allen mouse brain atlas is used.

This is an application paper. It makes use of existing technology (a Siamese network combined with a triplet or contrastive loss). Comparisons are made with finding the best matching plate by image registration using SimpleElastix and, in particular, by using mutual information after image registration as a ranking criterion. Experiments on real data show that the proposed metric learning approach significantly outperforms finding the best matching atlas plate via registration. Overall, the manuscript is well written and shows good results for the targeted task. Exploring the effects of artifacts, such as tilting errors, would have been interesting. Furthermore, it would have been interesting to see how much metric learning improves the results or if similar results could have been obtained by directly using an embedding from a pre-trained network (without further finetuning) as well as how much results are influenced by the mentioned data augmentation strategies.

---

### Official Review · Reviewer_hQtD · 2022-02-17

**Rating:** 3
**Confidence:** 4
**Recommendation:** Short Oral

**Deanonymize Review:**

no

**Detailed Comments:**

strengths:
- this is an interesting application paper with very challenging data
- the types of artefacts are interested and uncommon in other medical imaging domains


weaknesses:
- several parts of the paper could be streamlined and there are various redundant pieces of text.
- there is no methodological novelty, which is fine for an application paper.
- I would be interested how the other 4 slices in the Top-4 prediction would look like in Fig 4.? Are the close or randomly far away.

**Paper Type:**

validation / application paper

**Strengths Weaknesses:**

The paper is exploring the application of registering slices of mouse brain microscopy images to a common reference atlas. The type of images are up to 25600x20480 pixels. This data is channelling because of varying contrast and missing data. A Siamese network with contrastive loss is used to find correct locations for a given section in a reference atlas. Results are measured in Top 3 and Top 5 performance, which is between 80-100% accuracy.

strengths:
- interesting application
- difficult data
- reasonable results

weaknesses:
- no methodological novelty
- data is a bit limited (cross-population, clearly not re resolution)
- paper could be written more concisely

---

### Official Review · Reviewer_uWUV · 2022-02-20

**Rating:** 2
**Confidence:** 5

**Deanonymize Review:**

no

**Detailed Comments:**

A lot of work has been done over the last couple of decades on the problem of registering microscopy images. It would be nice for the authors to do a better job citing the existing literature and comparing or at least discuss how their method compares to these existing methods.

**Paper Type:**

methodological development

**Strengths Weaknesses:**

This paper proposes a deep learning algorithm to identify the closest atlas plate to a presented mouse brain microscopy image. The approach learns a low-dimensional space that captures the similarity between microscopy images and the reference atlas plates. The method uses CNNs, in a Siamese network configuration, to obtain low-dimensional embeddings of microscopy image data and atlas plates. These embeddings are contrasted with positive and negative examples in order to learn a semantically meaningful space that can be used for identifying corresponding 2D atlas plates. At inference, atlas plates that are closest to the microscopy image data in the learned embedding space are presented as candidates for registration.

Strengths:
The selected problem is challenging since microscopy images of brain slices often suffer from artifacts due to missing tissue regions, irregular staining, titling errors, air bubbles and tissue wrinkles. It has the potential to speed up and improve registration of 2D microscope images to an atlas.

Weakness:
This paper does not propose an image registration technique. Rather, it focuses on a method to select the best image in an atlas to register a given 2D slice. This is important, but it would have been nice for the authors to show how well their image selection improved the ultimate registration task.

The paper is not self-contained with respect to describing the two similarity cost functions in the Methods section. The reader must read [5] to understand the cost in Eq 1 and [13] to understand the cost function in Eq 2.

There is not very much experimental validation/evaluation of the method. 84 microscope images were used. It is not clear if all the datasets came from the same mouse or not. That is, where all these slices from one 3D data set of a mouse brain. The authors only compare their method to one other method. They compared their method in terms of accuracy and speed to SimpleElastix which used affine registration. It is not clear why this is a fair comparison since the proposed method selects the closest plate from the atlas, but does not do any registration.

---

### Official Review · Reviewer_LNeG · 2022-02-20

**Rating:** 4
**Confidence:** 4
**Recommendation:** Short Oral

**Deanonymize Review:**

no

**Detailed Comments:**

Considering that the paper does not propose novel methodology, but rather evaluates some existing approaches to solve a task of interest, my comments are mainly related to the conducted evaluation.

1. Dataset: How many brains/mice are included in the dataset? This may be relevant to assess robustness of the proposed approaches, and generality of the conclusions.

2. Ground Truth: With full understanding of the difficulty of the problem, I am anyway concerned that the way it is obtained may bias the results of the study. In Sect 3.1. It is stated that “the domain expert marked a single plate to be the ground truth depending on whichever plate best described specific regions of interest for their downstream task. This is to say, in most applications involving these data there are no hard ground truths…” How does a particular  “downstream task” affect the selection of one GT slice, and how is the “softness” of the GT incorporated in evaluation?
It would be highly beneficial, in my opinion, to evaluate the considered methods on an alternative - possibly even synthetic - dataset, with a more reliable GT, to get closer insight into the properties of the observed methods.

3. The results of the MAE metric, presented in Table 1 indicate rather similar performance of the evaluated methods - differences caused by the choice of backbone network architecture, learning strategy and loss function, as well as the level of dimensionality reduction are very small, and do not support any strong conclusions regarding the pros and cons, at least on the observed dataset. Considering the comparably huge difference in performance of the observed base-line (Simple Elastix), this calls for both further analysis of the learning based approaches (on e.g. other datasets), and for re-evaluation of the suitability of the selected base-line.


4. Please provide more information about the SimpleElastix configuration. Which are the parameters that you have tuned and how? What are the final values that are used to produce the reported performance? Why is Mutual Information Maximization considered to be most suitable for the task, when SimpleElastix is used? Being a natural objective for multi-modal image registration, it is a less straightforward choice for mono-modal cases. Are other similarity/distance functions considered?



5. Table 2 clearly shows superiority of the proposed approaches to the approach based on residual distance after MI maximization. Some comments on the outperformed approach may be of interest. Potentially, a Top-1 accuracy might be illustrative, in case that SimpleElastix actually manages to detect the correct slice in 25% of the cases. This will not affect the main conclusion of the study, but may add some relevant informative content. In general, it will be useful for the reader to get a better understanding on why the baseline approach reaches such poor performance, e.g., description/illustrations of the most typical modes of failure.


6. I suggest to take a look at a recently published study “Cross-Modality Sub-Image Retrieval using Contrastive Multimodal Image Representations”, E Breznik, E Wetzer, J Lindblad, N Sladoje, https://arxiv.org/abs/2201.03597
It addresses a closely related problem, and evaluates several strategies, potentially providing relevant  reference performance for this paper.


**Paper Type:**

validation / application paper

**Strengths Weaknesses:**

Summary:

The paper focuses on the first step within atlas based registration of microscopy images of slices of a mouse brain - identification of which 2D microscopy image in an atlas, to be used as a reference image to register to. A correctly selected reference image is crucial for success of subsequent analysis results, whereas problems arise due to difference in cutting angles, different image sizes/fields of view, and a range of artifacts resulting from the imaging. Furthermore, the size of the observed images prevents exhaustive comparisons with the whole set of images in the atlas. The identification is therefore often performed manually by experts, calling for suitable methods that can overcome the listed challenges and allow for reliable automated identification.
Authors suggest applying learning-based techniques to extract low dimensional image representations of both the atlas images and the queries, and to efficiently compare the representations, instead of the original images. Due to achieved reduction of data, the comparisons can be performed in orders of magnitude shorter time, while reaching superior Top-5 accuracy of identification, compared to what is achieved by relying on residual distance after registration by Mutual Information maximization using SimpleElastix.
The suggested representation learning methods (relying on Siamese network configurations)  are all well established, and the paper does not introduce any novelty in that respect. At the same time, they are suitably chosen for the task. Evaluation is limited (only one relatively small dataset is used, MI maximization is SimpleElastix is not a convincing reasonable base-line) but clearly shows that the suggested direction is promising.
The code is available, supporting reproducibility. The dataset, however, seems to be private, which somewhat limits the impact of the study, especially considering that this is (as mentioned) the only dataset considered in the paper.




Strengths:
- The study is well motivated and addresses a relevant problem.

- The suggested methods to address the identified problem perform well on the task and outperform the (reasonably selected) alternative approach.

- The paper is well written, with a good structure.

- The paper fits well into the scope of the workshop and is expected to be of interest for its audience, and informative for other readers facing a task of retrieval of large images.



Weaknesses:
Weaknesses are mainly related to, in my opinion, limited evaluation.

- Evaluation is performed on only one private dataset. This limits confidence in assessment  of the impact of this study on other similar problems. Furthermore, Ground Truth appears to be rather unreliable, which may affect the drawn conclusions.

- The dataset used in the study is rather small, which limits the reliability of the conclusions.

- The obtained results indicate that the general approach to learn image representations and compare them to identify matches in the atlas to perform registration to is very promising, in particular in comparison with the selected reference method. However, further analysis of the results presented in e.g. Table 1, is required to understand advantages and disadvantages of the individual considered learning strategies.

- Comparison with (only) MI maximization with SimpleElastix as the baseline method is, in my opinion, too limited. Image retrieval is an established problem and I would appreciate at least some comments on the existing methods from this field.

---

### Decision · Program_Chairs · 2022-02-22

Accept